# Magnetosheath jet properties and evolution as determined by a global hybrid-Vlasov simulation

Minna Palmroth[1,2], Heli Hietala[3,4], Ferdinand Plaschke[5,6], Martin Archer[7,8], Tomas Karlsson[9], Xóchitl Blanco-Cano[10], David Sibeck[11], Primož Kajdič[10], Urs Ganse[1], Yann Pfau-Kempf[1], Markus Battarbee[1], and Lucile Turc[1]

[1]Department of Physics, University of Helsinki, Helsinki, Finland
[2]Space and Earth Observation Centre, Finnish Meteorological Institute, Helsinki, Finland
[3]Department of Physics and Astronomy, University of Turku, Finland
[4]Department of Earth, Planetary, and Space Sciences, University of California, Los Angeles, USA.
[5]Institute of Physics, University of Graz, Graz, Austria.
[6]Space Research Institute, Austrian Academy of Sciences, Graz, Austria.
[7]The Blackett Laboratory, Imperial College London, London, UK
[8]School of Physics and Astronomy, Queen Mary University of London, London, UK
[9]School of electrical engineering and computer science, KTH Royal Institute of Technology, Stockholm, Sweden.
[10]Instituto de Geofísica, Universidad Nacional Autónoma de México, Mexico City, Mexico
[11]Code 674, NASA/GSFC, Greenbelt, MD, USA

*Correspondence to:* Minna Palmroth (minna.palmroth@helsinki.fi)

**Abstract.** We use a global hybrid-Vlasov simulation for the magnetosphere, Vlasiator, to investigate magnetosheath high-speed jets. Unlike many other hybrid-kinetic simulations, Vlasiator includes an unscaled geomagnetic dipole, indicating that the simulation spatial and temporal dimensions can be given in SI units without scaling. Thus, for the first time, this allows investigating the magnetosheath jet properties and comparing them directly with the observed jets within the Earth's magnetosheath. In the run shown in this paper, the interplanetary magnetic field (IMF) cone angle is 30°, and a foreshock develops upstream of the quasi-parallel magnetosheath. We visually detect a structure with high dynamic pressure propagating from the bow shock through the magnetosheath. The structure is confirmed as a jet using three different criteria, which have been adopted in previous observational studies. We compare these criteria against the simulation results. We find that the magnetosheath jet is an elongated structure extending earthward from the bow shock by ∼2.6 $R_E$, while its size perpendicular to the direction of propagation is ∼0.5 $R_E$. We also investigate the jet evolution, and find that the jet originates due to the interaction of the bow shock with a high dynamic pressure structure that reproduces observational features associated with a short, large-amplitude magnetic structure (SLAMS). The simulation shows that magnetosheath jets can develop also under steady IMF, as inferred by observational studies. To our knowledge, this paper therefore shows the first global kinetic simulation of a magnetosheath jet, which is in accordance with three observational jet criteria, and is caused by a SLAMS advecting towards the bow shock.

# 1 Introduction

Earth's magnetosphere is surrounded by the magnetosheath, which consists of shocked and turbulent plasma of solar wind origin. The sunward boundary of this region is the bow shock through which the solar wind plasma flows into the magnetosheath. The earthward boundary of the magnetosheath is the magnetopause, the outer edge of Earth's magnetosphere. The bow shock and magnetosheath plasma properties relative to those in the upstream pristine solar wind depend broadly on the interplanetary magnetic field (IMF) direction. One of the most important defining conditions within the magnetosheath is the angle between the bow shock normal and the IMF. In portions of the bow shock, where the bow shock normal lie more or less parallel to the IMF direction, the bow shock is said to be *quasi-parallel*. At the quasi-parallel shock, part of the solar wind particles reflect back towards the Sun (Schwartz et al., 1983; Meziane et al., 2004), causing instabilities and waves upstream, and forming a so-called foreshock. The region downstream from the quasi-parallel shock is called the quasi-parallel magnetosheath, where the plasma properties are highly turbulent (e.g., Fuselier et al. 1991; Gutynska et al. 2012). On the other hand, the region downstream from the quasi-perpendicular side is less turbulent. There is no foreshock upstream from the quasi-perpendicular bow shock because IMF lines keep the reflected particles close to the bow shock and the waves do not have time to grow. Nevertheless, the magnetosheath downstream from the quasi-perpendicular bow shock hosts a variety of locally-generated waves, e.g., mirror mode waves (Soucek et al., 2015; Hoilijoki et al., 2016).

Němeček et al. (1998) reported observations of peaks in the ion fluxes within the quasi-parallel magnetosheath, which they termed transient flux enhancements. Several studies have since investigated the properties of these high-speed structures that have been termed as magnetosheath jets, and demonstrated their importance in terms of geoefficiency. They can for example distort the magnetopause (Shue et al., 2009; Plaschke et al., 2016), and drive magnetospheric dynamics because they can trigger magnetopause reconnection (Hietala et al., 2018). Statistical investigations of the jets show that they are clearly associated with the foreshock and the quasi-parallel magnetosheath (e.g., Archer and Horbury 2013; Plaschke et al. 2013). Omidi et al. (2016) therefore suggested that foreshock waves may be related to the origin of the jets. Hietala et al. (2009) proposed a mechanism to produce the jets by a rippled bow shock, which collimates particles into a high-speed structure. Karlsson et al. (2015) suggested that the jets could be associated with foreshock short, large amplitude magnetic structures (SLAMS, Lucek et al. 2002, 2004) originating from steepening foreshock waves, and travelling through the bow shock.

Originally, the jets were observationally identified by high velocities (e.g., Němeček et al., 1998; Hietala et al., 2009). In recent years the vast majority of observational studies have used dynamic pressure and not velocity as the key quantity, although a level of agreement is expected due to the quadratic dependence of the velocity in the dynamic pressure. Plaschke et al. (2013) devised a criterion $C_P$, defined as the ratio of the magnetosheath dynamic pressure in the $X$ direction to the upstream solar wind dynamic pressure. Plaschke et al. (2013) defined that in order to represent jets, $C_P$ had to fulfil the condition

$$C_P = \frac{\rho v_X^2}{\rho_{sw} v_{sw}^2} > 0.25, \tag{1}$$

where $\rho$ is the density, $v_X$ is the velocity component in the $-X$ direction, the numerator refers to the conditions in the magnetosheath while the denominator represents solar wind conditions, with the subscript $sw$ denoting the solar wind. The

coordinate system that they used was Geocentric Solar Ecliptic (GSE), where $X$ is sunward, $Z$ is perpendicular to the ecliptic plane and is positive northward, and $Y$ completes the righthanded system. The Plaschke criterion $C_P$ defines the jet as the entire region where Eq. (1) holds, and requires that the dynamic pressure peak is >0.5 times the solar wind value. Further, the criterion is applied only for solar zenith angles less than $30°$.

Archer and Horbury (2013) used the total dynamic pressure but divided by a 20-minute temporal average of the dynamic pressure within the surrounding magnetosheath, and required that

$$C_A = \frac{\rho v^2}{<\rho_{sh} v_{sh}^2>_{20min}} > 2. \tag{2}$$

where the brackets indicate a temporal average. Karlsson et al. (2012) investigated enhancements in the magnetosheath density, which they called plasmoids. They separated the plasmoids according to their speed, and remarked that the fast plasmoids

whose local velocity increased at least 10% could be associated with jets. They defined the events by taking ratios of the magnetosheath electron density to a 15-minute temporal average within the magnetosheath as

$$C_K = \frac{n_e}{<n_e>_{15min}} > 1.5, \tag{3}$$

where $n_e$ is the electron density in the magnetosheath. Both $C_A$ and $C_K$ are only defined to identify peak values of the relevant parameters, and when durations or spatial scales were identified the full-width-at-half-maximum was used. Jets identified with

the three criteria are in broad agreement with respect to occurrence and properties, suggesting that the criteria identify similar phenomena. This motivates a modelling study to test how similar the three criteria in fact are and whether they all are associated with magnetosheath jets.

Hao et al. (2016) performed local hybrid-particle-in-cell (PIC) simulations within a limited spatial extent, and found that the solar wind Alfvén Mach number is important in determining how far the jets can penetrate within the magnetosheath. Using

a 2D hybrid-PIC code, Karimabadi et al. (2014) observed elongated structures with higher magnetic field and plasma density traversing from the foreshock to the magnetosheath. However, since Karimabadi et al. (2014) use a scaled dipole strength in the hybrid-PIC model, representative of roughly a Mercury-sized magnetosphere, deducing the scale sizes of the structures from the simulation results is not straightforward, and their direct comparison to the jets observed in the Earth's magnetosheath is difficult. Nevertheless, Karimabadi et al. (2014) reported that jet scales parallel to the direction of propagation could be $\sim$2.4

$R_E$, and in the perpendicular direction $\sim$0.3 $R_E$ within the Earth's magnetosheath. Observationally, Plaschke et al. (2016) estimate that the characteristic jet sizes are 1.34 $R_E$ by 0.71 $R_E$.

This paper employs the hybrid-Vlasov simulation code Vlasiator to investigate the jet properties. Vlasiator includes ion-kinetic features similar to hybrid-PIC codes, but unlike hybrid-PIC codes, does not include sampling noise in the results due to a different modelling approach. Further, Vlasiator uses the actual unscaled geomagnetic dipole strength as a boundary

condition, and therefore the results can be given in $R_E$ and seconds without scaling, indicating that the length and time scales can be directly compared to spacecraft observations of jets. In this paper, we first introduce Vlasiator, and the run used to

examine the magnetosheath jets. We visually identify a candidate jet, after which we show that our candidate jet fulfils all three jet criteria described above. We then examine the jet properties and evolution, and analyse the process that generates the jet, before ending the paper with discussion and conclusions.

## 2 Model

Vlasiator is a hybrid-Vlasov model for global simulations of the Earth's magnetosphere. Vlasiator solvers treat protons as a distribution function $f(\mathbf{r}, \mathbf{v}, t)$ in phase space, and electrons as a massless charge-neutralizing fluid (Palmroth et al., 2013; von Alfthan et al., 2014; Palmroth et al., 2015; Pfau-Kempf, 2016). Electron kinetic effects are neglected by the solvers, but the ion kinetic effects are solved without numerical noise. The time-evolution of $f(\mathbf{r}, \mathbf{v}, t)$ is controlled by the Vlasov equation, propagated by a fifth-order accurate semi-Lagrangian approach (Zerroukat and Allen, 2012; White and Adcroft, 2008). Maxwell's equations neglecting the displacement current in the Ampère-Maxwell law are used to solve the electromagnetic fields. Maxwell's equations are supplemented by Ohm's law, including the Hall term. The technical features of the code including the closure scheme, the numerical approach, and the parallelization techniques are described by von Alfthan et al. (2014) in the previous version using the Finite Volume Method, while here and in Palmroth et al. (2015) an updated Semi-Lagrangian scheme is used (see also Pfau-Kempf 2016).

The run is carried out in the ecliptic $XY$ plane of the Geocentric Solar Ecliptic (GSE) coordinate system, representing a two-dimensional (2D) approach in ordinary space. Each ordinary space simulation cell includes a 3D velocity space used to describe the proton velocity distribution. Therefore the approach here is 2D-3V in total. The simulation plane in the run used in this paper ranges from $-7.9$ $R_E$ to 46.8 $R_E$ in $X$, and $\pm 31.3$ $R_E$ in $Y$, with a resolution of 228 km corresponding to the typical ion inertial length in the solar wind. The velocity space resolution is 30 km/s. The solar wind parameters are given as an input at the sunward wall of the simulation box, while copy conditions are applied at other boundaries. The $Z$ direction in ordinary space applies periodic conditions. The inner edge of the magnetospheric domain is a circle with a radius of 5 $R_E$, while the ionosphere is a perfect conductor in the present version of the code. The same run has also been used to examine magnetosheath mirror mode waves by Hoilijoki et al. (2016), with a general agreement to existing knowledge of the phenomenon.

The solar wind parameters in this run are as follows: Solar wind distribution functions are assumed Maxwellian, with an initial temperature of 0.5 MK. The IMF has a cone angle of $30°$, the IMF $x$ component is $-4.33$ nT, IMF $y$ is 2.5 nT, while the total magnetic field intensity is 5 nT. The solar wind density is 1 cm$^{-3}$, and velocity 750 kms$^{-1}$ in the $-X$ direction. The combination of the solar wind parameters have been chosen to facilitate on one hand the relatively fast initialisation of the simulation to save in the total computational load, and on the other hand the realistic representation of the foreshock. With these solar wind parameters, the upstream Alfvén Mach number becomes 7, well inside the normal range of Alfvén Mach numbers at the Earth (Winterhalter and Kivelson, 1988). Thus we can trust the foreshock physics and consequently its interactions with the bow shock. The combination of solar wind values yields a relatively low dynamic pressure of about 1 nPa, however, this dynamic pressure or lower are observed about 23% of the time under quasi-radial IMF throughout the solar cycle, based on OMNI solar wind data. Observational statistics show a slight tendency for jets to occur for higher solar wind speeds and lower

densities than usual (Plaschke et al., 2018), indicating that our solar wind parameter set represents the conditions under which the magnetosheath jets occur.

Before going to the results, we note that the bow shock moves gradually upstream in in all 2D hybrid-kinetic models. There are two reasons for this. First, the magnetosheath magnetic field piles up in front of the magnetopause because in 2D it cannot slip around the magnetosphere towards the nightside as in reality. Secondly, there is an artificial heating in the hybrid-kinetic simulations due to numerical diffusion. This feature is relatively minor in Vlasiator, and the numerical heating does not contribute significantly to the gradual expansion of the bow shock (e.g., von Alfthan et al., 2014; Palmroth et al., 2015).

## 3 Results

Figure 1a shows a close-up of the Vlasiator simulation domain investigated in this paper. It shows a snapshot of a supplementary movie S1, depicting the dynamic pressure at time $t = 305.5$ s from the beginning of the run. Colour-coding shows the dynamic pressure. To guide the eye, Fig. 1a also includes the bow shock position as a white solid line, depicting the location where the density is twice the solar wind density. The shock compression ratio is about $3-4$ at Earth, making the density gradient at the shock quite sharp, and therefore the bow shock position can be shown in this simple manner. As for the magnetopause position, we first note that in this 2D-3V simulation it is not realistic to expect that the magnetopause position agrees exactly with the empirical proxies. Further, in magnetohydrodynamic (MHD) simulations, such as in GUMICS-4 (Janhunen et al., 2012), the location of the magnetopause depends upon the parameter by which it is defined. The so called fluopause, determined by an average of solar wind streamlines deflecting around the magnetosphere, is a good proxy for the magnetopause, well in accordance with empirical proxies (Palmroth et al., 2003). Therefore we also show the streamlines in Figure 1a to illustrate roughly the dimensions of the magnetosheath in this run. Following Palmroth et al. (2003), the subsolar magnetopause would be determined by neglecting the innermost streamline at around 7 $R_E$, and by taking an average of the next ones towards upstream, placing the magnetopause using this proxy to somewhere around 10 $R_E$.

Based on movie S1, we visually identified a high-pressure structure emerging from the bow shock surface and extending through the magnetosheath, marked with a white arrow in Fig. 1a. The supplementary movie S1 shows both the beginning and the end of the visually identified feature. As we shall describe in this paper, the feature is associated with a higher dynamic pressure advecting towards the bow shock, and reaching it at around $t = 282$ s. On the other hand, at $t = 325 - 340$ s, the visually identified feature seems to be associated with a transient wave or an oscillation, which originates approximately at $X, Y = [7.5, -4]$. This transient follows from the arrival of the remnant of the visually identified feature, and two pulses traveling away from the impact point are visible. In a 2D-3V simulation, we do not wish to confirm whether features close to the magnetopause are realistic due to the pile-up effect described above. However, from Supplementary movie S1 it is clear that the visually identified feature is certainly a transient event having a distinct lifetime. It has such a large dynamic pressure that it pushes ambient plasma and has an impact downstream. Therefore we take this feature into a closer scrutiny in order to conclude about its relevance to the magnetosheath jets.

The white dot in Fig. 1a at $X, Y = [9.5, -4.2]R_E$. shows the earthward edge of this structure, from which we show virtual spacecraft data in Fig. 1b. The virtual spacecraft data in Fig. 1b shows that the velocity increased roughly by 20%, density roughly by 50%, while the dynamic pressure roughly doubled at the time of the structure in panel 1a, marked by a vertical dashed line.

Figure 2 shows the Plaschke criterion $C_P$ defined in Eq. (1) in a spatially limited zoom of Fig. 1. The colour-coding shows the dynamic pressure ratio between the magnetosheath and solar wind, using the $X$ component of the velocity $v_X$. The black contour shows where this quantity exceeds 0.25, while the white contour shows the area where the quantity exceeds 0.5 in line with Plaschke et al. (2013). The structure in Fig. 1 can be observed as an elongated feature starting from the bow shock and extending to the left towards the magnetopause in Fig. 2 approximately at $X, Y = [10, -4]R_E$.

Using the same zoom as Fig. 2, Fig. 3 shows the Archer and Horbury criterion $C_A$ (Eq. 2), which is a ratio of the dynamic pressure and the temporal average of dynamic pressure. Panel 3a shows this ratio, panel 3b presents the dynamic pressure (the numerator of the criterion), and panel 3c shows the temporal average of dynamic pressure (the denominator of the criterion). While Archer and Horbury (2013) originally used a 20-minute average in the denominator, here we use a three-minute temporal average, centered on time $t = 305.5$ s. This is solely because the simulation interval does not last for 20 minutes, and while

testing different values this three-minute average was found to be the shortest period identifying the structure, while having a manageable amount of data. The contours in panel 3a show where the Archer and Horbury criterion exceeds 2, and where therefore the dynamic pressure is twice the temporal average. The largest area satisfying this criterion can be found near the location $X, Y = [9, -4]R_E$.

     Figure 4 shows the Karlsson criterion $C_K$ (Eq. 3), namely the ratio of the instantaneous density to the temporal average of

20 the density. Panel 4a shows this ratio. Panels 4b and 4c show the density and the temporal average of density over three minutes, centered on the time $t = 305.5$ s, respectively. The contour in panel 4a shows locations where the ratio exceeds 1.5., that is where the density is 50% greater than the temporal average. Figure 4a shows that the Karlsson criterion is fulfilled mostly at the surface of the bow shock, while a small area of higher density can be found at location $X, Y = [9, -4]R_E$.

     Finally, Fig. 5 compares results for all the criteria, the Karlsson criterion $C_K$ in Eq. (3) with magenta, Archer and Horbury

criterion $C_A$ in Eq. (2) with blue, and the Plaschke criterion in Eq. (1) with a black contour. The region we visually identified from the movie S1, and which is indicated by an arrow in Fig. 1a fulfils all three criteria approximately at $X, Y = [10, -4]R_E$. Since the criteria agree, we call the feature a magnetosheath jet, and identify its physical dimensions and evolution in time. We adopt an inclusive strategy, and determine that the jet originates at the bow shock with enhanced $C_K$ (magenta) criterion, at $X = 11.6 R_E$, and reaches a location with enhanced $C_A$ (blue) criterion at $X = 9.1 R_E$. Taking into account the angle at

which the magnetosheath jet propagates from the bow shock towards the magnetopause, its length is approximately 2.6 $R_E$ in the direction of propagation. In the perpendicular direction, the jet size varies from 0.6 $R_E$ at the bow shock, to 0.3 $R_E$ in the mid-jet area, to ~0.5 $R_E$ at the magnetopause end. Since Fig. 5 represents a snapshot, we emphasise that these dimensions are instantaneous values.

     Next we investigate the evolution of the jet size in time in Fig. 6, continuing with the inclusive strategy. The panels of Fig.

6 present the jet area, radial size, and tangential size, respectively. The area has been calculated such that both the Archer

and Horbury as well as the Plaschke criteria delimit the jet, and the area is the sum of the areas of the grid cells within the jet boundaries. The radial size is simply the subtraction of the maximum and minimum radial distance of the jet boundary positions, while the tangential size is the jet area divided by the radial distance. Figure 6 indicates that the area increases and decreases during the jet lifetime, and reaches its maximum just before the time of the jet in Fig. 5. The radial size increases first as the jet emerges from the bow shock, but then stays constant as it propagates through the magnetosheath before the jet disperses away. The tangential size remains below 1 $R_E$ on average for the most part of the jet lifetime, but the increase of the tangential size at the end of the jet lifetime suggests that it disperses into the tangential direction.

Figure 7 investigates how the jet profile changes as a function of distance from the bow shock. Figure 7a shows an overview plot, with both Plaschke, and Archer and Horbury criteria used to delimit the jet. Figure 7a shows three coloured stars in positions: green $= [9.2, -3.7]$, red $= [10.0, -4.4]$, cyan $= [10.8, -5.2]$. Figure 7b shows velocity, density, and dynamic pressure as a function of time at these three locations, with similar colour-coding as the stars are given in panel 7a. The full-width-at-half-maximum, which would be measured by a spacecraft, changes from 14 s to 8 s and 9 s from the bow shock to the mid-jet, and to the earthward tip, respectively. Converting these to spatial scales with multiplying with the average velocity yields a spatial size of 0.7 $R_E$ $-$ 0.3 $R_E$, respectively. Clearly, the velocities and the dynamic pressures are greatest nearest the shock, and decrease as the jet propagates towards the magnetopause. The dynamic pressure decreases by 70% from the bow shock to the vicinity of the magnetopause, indicating that the origin of the jet may be related to the dynamic pressure outside the bow shock.

Figure 8 examines what causes the jet, using the Plaschke criterion. Figures 8a-d show the total dynamic pressure in the background, and the Plaschke criterion as a black contour at four times near the time shown in Fig. 5. The panels are snapshots from Supplementary movie S2. In Fig. 8a, a high-pressure area shown by the white arrow approaches the bow shock. This high-pressure structure steepens towards the bow shock surface within a matter of seconds. At time $t = 295$ s the structure has hit the bow shock, shown by the arrow in Fig. 8b. In panels 8c and 8d this bulge extends towards the magnetopause, and at time $t = 310$ s it is already fading away. Supplementary movie S2 shows this time sequence in a more dynamic fashion.

Finally, we investigate the high-pressure structure that causes the jet in more detail. Figure 9a shows the high-pressure feature advecting towards the bow shock with the solar wind. The black dot near the centre of the high-pressure structure shows a point at which we take virtual spacecraft data in Fig. 9b. The parameters in Fig. 9b are chosen to facilitate a comparison to a SLAMS, which shows an increase in the magnetic field by a factor of 2 or more, and contains a rotation of the magnetic field vector (Lucek et al., 2002, 2004). Figure 9b shows a twofold increase of both the density and the magnetic field intensity when the structure passes the virtual spacecraft location. The components of the magnetic field indicate that the structure includes a clear rotation in the $XZ$ plane. Therefore we conclude that the high-pressure structure that causes the jet reproduces the observational criteria (Lucek et al., 2002, 2004), suggesting that it is indeed a SLAMS.

## 4 Discussion

We have presented a Vlasiator simulation run in the ecliptic plane with a 30° IMF cone angle. We identify and study a magnetosheath jet, and verify its properties by comparing them to three observational criteria (Plaschke et al., 2013; Archer and Horbury, 2013; Karlsson et al., 2012, 2015). The fact that the structure we observed fulfilled all three observational criteria
indicates that the observations of Plaschke et al. (2013); Archer and Horbury (2013); Karlsson et al. (2015) indeed concern similar phenomena within the magnetosheath. The fact most supporting the idea that our visually selected event is indeed a magnetosheath jet is that all three criteria agree spatially within the jet, and that the identified region is continuous starting from the shock surface and reaching towards the magnetopause. Further, it has a limited lifetime during which the criteria are met within the same region, suggesting that the origin has to do with temporal changes that are connected by the three
criteria. While we have concentrated on one jet, there are many more candidate jets in this Vlasiator run that satisfy the different criteria, as shown by the Supplementary movies S1 and S2. This and other runs carried out with Vlasiator will allow statistical investigations looking into the evolution of the jets as a function of their position within the magnetosheath, their size distribution, and how these parameters depend on the driving conditions.

    We find that the jet size in the direction of propagation is at maximum 2.6 $R_E$, while in the perpendicular direction it is $\sim$0.5
$R_E$ in size. These dimensions are in agreement with previous scaled results given in ion inertial lengths within a hybrid-PIC simulation with roughly a Mercury-size magnetic dipole, assuming typical magnetosheath properties in order to convert the results into Earth radii (Karimabadi et al., 2014). Plaschke et al. (2016) estimate the characteristic size of the jets to be 1.34 $R_E$ by 0.71 $R_E$, while the jet in this paper is within the range of the jet sizes reported by them. Contrary to observations, in the simulation the entire jet can be measured and the flow parallel direction can be identified. Spacecraft will rarely cross the
jet along the axis of largest extent. Thus an exact match between observationally identified and modelled jets are not to be expected, but the fact that they broadly agree suggest that the modelled jet can be examined in more detail, and conclusions about its properties can be related to the observations.

    It is interesting to compare the different observational criteria in Eqs. (1-3) in light of the simulation results shown here. According to Plaschke et al. (2018), the Archer and Horbury criterion is most inclusive, identifying the largest number of jets,
while the Karlsson criterion is most strict identifying the smallest number of jets (or plasmoids). We have not rigorously tested how large areas the three criteria in fact concern within the magnetosheath, as we have concentrated on finding a structure that could be identified as a jet with the present observational criteria. We note however that based on Fig. 5 both the Archer and Horbury (2013) and the Plaschke et al. (2013) criteria identify larger regions than the Karlsson criterion, which indeed seems to be the most strict in the simulation overall. It is also interesting to note that while the widely accepted term "jet"
has a connotation of an elongated feature, according to the results shown here, the Archer and Horbury (2013) and Karlsson et al. (2012, 2015) criteria delineate features shaped more like "blobs". Without vast fleets of observing satellites, it falls on a combination of observational and simulational efforts to infer the shapes and dimensions of jets. Further modelling studies of the jet size distributions will be necessary in order to assess this point.

We find that the Karlsson criterion is mostly fulfilled near the bow shock surface, and it seldom reaches the magnetosheath portions close to the magnetopause. On the contrary, the Archer and Horbury (2013) criterion identifies regions closest to the magnetopause but can be found to be satisfied throughout the magnetosheath, agreeing with the observational statistics. These characteristics might be associated with the solar wind driving conditions in our run. Neither Karlsson et al. (2012) nor Karlsson et al. (2015) specify the solar wind conditions for their events, while our event is associated with a solar wind density of $1\,\text{cm}^{-3}$. The Archer and Horbury criterion is determined by the dynamic pressure, which depends on the square of the velocity, which in our simulation is rather high in the solar wind, $750\,\text{kms}^{-1}$. While both criteria concern ratios that can be enhanced during a variety of driving conditions, it is possible that in the conditions of this run, the Karlsson high-density plasmoids are either not properly generated or cannot propagate deep in the magnetosheath, while the Archer and Horbury pressure enhancements could traverse further towards the magnetopause due to the faster general velocities in the magnetosheath. In accordance with Plaschke et al. (2013), the Plaschke criterion in our results is most enhanced near the bow shock. This may be because it is based on the $X$ component of dynamic pressure: The general magnetosheath flow pattern starts to deviate from the $X$ direction near the shock. Further, the jets push ambient magnetosheath plasma out of their way in order to reach the magnetopause, decelerating them to level that no longer satisfies the Plaschke criterion. As we also show that the dynamic pressure rapidly decreases as a function of distance from the bow shock, to observe jets closer to the magnetopause it may be better to choose the Archer and Horbury (2013) criterion.

Both ULF waves and SLAMS are common in the foreshock, where they advect towards the bow shock (e.g. Eastwood et al., 2005, and references therein). By looking at the Supplementary movie S2 and Figs. 8 and 9, we find that the jet in question is formed by the interaction of a high pressure structure with the bow shock. The pressure enhancement has a larger pressure than its neighbours, and it is elongated along the $X$ axis, and wider in $Y$ than other foreshock fluctuations within the run sequence. Based on virtual spacecraft data taken from the structure, we conclude that its characteristics reproduce the main features of a SLAMS. As the bow shock already shows an initial dent before the SLAMS arrives, the SLAMS can pass the bow shock with little braking and can propagate deep into the magnetosheath. In contrast, we refer to another larger pressure fluctuation that reaches the bow shock at about $t = 351$ s (at $X, Y \approx [11, -3.5]R_E$, see movie S2). The bow shock is not dented upon the arrival of this fluctuation and therefore the resulting jet-like structure does not grow large or propagate very deep within the magnetosheath.

Omidi et al. (2016) used a 2D hybrid-PIC simulation to associate magnetosheath jet-like structures with foreshock ULF waves. The jets reported by Omidi et al. (2016) almost reach the magnetopause, and they are associated with high dynamic pressures. The authors note that "these regions are not associated with high flow speeds and are instead caused by the density enhancements associated with the magnetosheath filamentary structures". Without a rigorous comparison to the data in Omidi et al. (2016) we cannot be sure that the features in their simulation and the ones shown here concern the same physics and whether therefore the origins of the structures can be related. However, we do note that in our simulations the higher dynamic pressure regions within the magnetosheath, which we call the magnetosheath jets, are associated with high velocities. Further, Hao et al. (2016) carried out a local 2D hybrid-PIC simulation with a planar shock to investigate a jet-like feature. They associated the jet-like feature with the upstream ULF waves, and made a note that it may originate due to a "SLAMS-like"

feature interacting with the bow shock. The present study takes further these previous numerical works by providing a global simulation of the formation and evolution of magnetosheath jets in the real magnetospheric scales, directly comparable to those observed by Earth-orbiting spacecraft. This allows us to rigorously compare the jet with existing observational criteria, and also to identify the structure causing the jet as a SLAMS. To our knowledge, this is the first time this type of study has been carried out.

As for the generation of the jets, Hietala et al. (2009) suggested a mechanism, which relies on an assumption of a rippled shock surface that actively funnels particles into a collimated structure having a high velocity, propagating towards the magnetopause. Hietala et al. (2009) discussed the origins of such a ripple and remarked that while rippling is inherent to the quasi-parallel shock, one possible origin for the ripple would be a SLAMS convecting towards the bow shock and interacting with it. In contrast, Karlsson et al. (2015) suggested that foreshock SLAMS could essentially travel through the bow shock and maintain its higher pressure, if there is an original dent or corrugation at the bow shock surface to which that SLAMS hits. The jet generation we have investigated here is directly associated with a SLAMS coming into a contact with a dented bow shock, after which that SLAMS essentially continues through the magnetosheath as a structure that resembles a jet, which fulfils the jet observational criteria. Therefore our results confirm the Karlsson et al. (2015) scenario for this single jet. However, this does not rule out other possible generation mechanisms that may also be in action.

## 5 Conclusions

We investigated magnetosheath high-speed jets in a hybrid -Vlasov simulation done at scales directly comparable to the Earth's magnetosphere. We identify structures in the simulation that can be related to the magnetosheath jets using three different observational criteria. We examine one such jet in more detail and find that its maximum size is 2.6 $R_E$ and $\sim$0.5 $R_E$ in the direction parallel and perpendicular to the propagation direction, respectively. The jet is caused by a SLAMS structure travelling through the bow shock.

*Acknowledgements.* We acknowledge The European Research Council for Starting grant 200141-QuESpace, with which Vlasiator (http: //physics.helsinki.fi/vlasiator) was developed, and Consolidator grant 682068-PRESTISSIMO awarded to further develop Vlasiator and use it for scientific investigations. We gratefully also acknowledge the Finnish Centre of Excellence in Research of Sustainable Space (Academy of Finland grant number 312351), Academy of Finland grant numbers 267144, and 309937. PK's work was supported by DGAPA/PAPIIT grant IA104416. The CSC − IT Center for Science in Finland is acknowledged for the Grand Challenge award leading to the results shown in here. We acknowledge valuable discussions within the International Space Science Institute (ISSI) team 350, called "Jets downstream of collisionless shocks", led by FP and HH. LT acknowledges Marie Sklodowska-Curie grant 704681. HH was supported by the Turku Collegium for Science and Medicine, and NASA NNX17AI45G. We thank Mr Jonas Suni for producing data for the figures in the revised version.

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

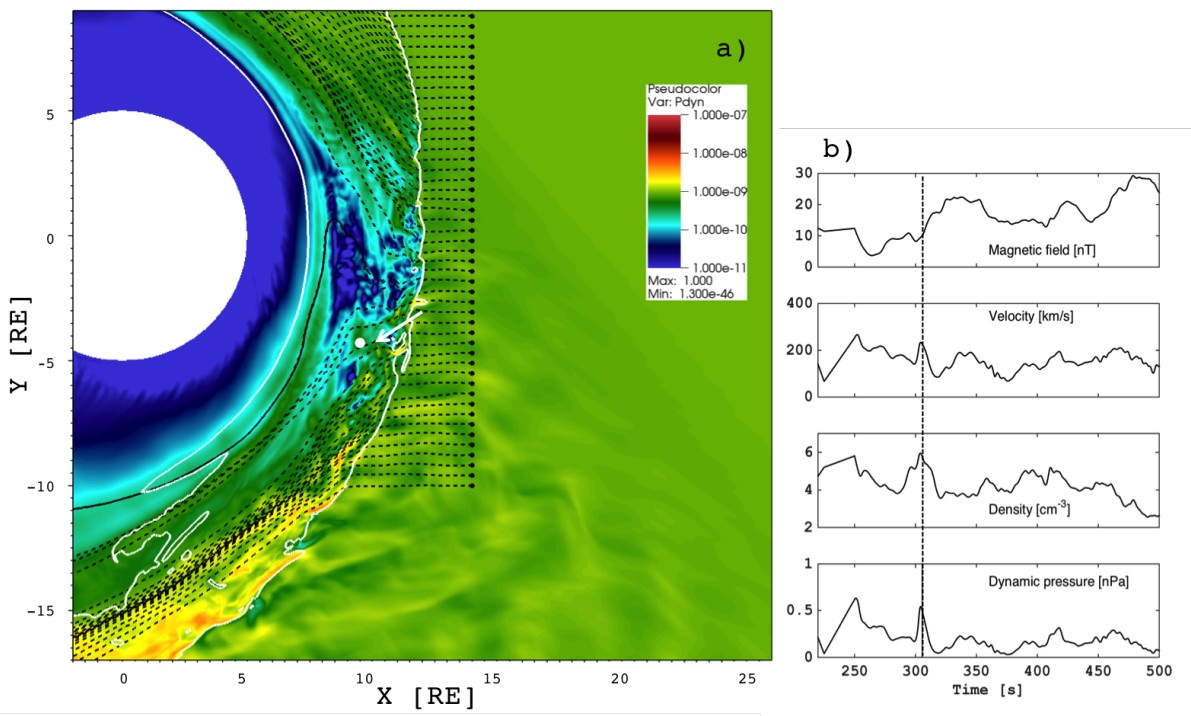

**Figure 1.** a) Dynamic pressure within Vlasiator simulation domain. Bow shock position is identified with white solid line. The black dashed lines are solar wind streamlines illustrating the magnetopause position roughly (see text for details). The figure is a snapshot of Supplementary movie S1, which does not include the bow shock position or the streamlines. The arrow indicates the visually detected magnetosheath jet under scrutiny in this paper. b) Virtual spacecraft data from the location marked with a white dot in panel a): Magnetic field, velocity, density and dynamic pressure as a function of time. The dashed vertical line shows the time of the visually identified jet in panel a).

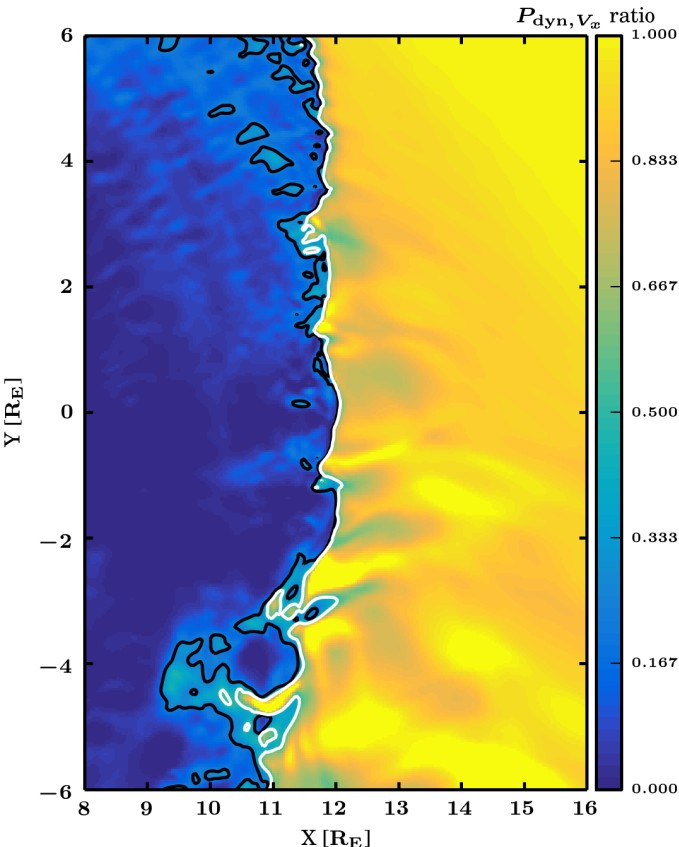

**Figure 2.** Colour-coding shows the dynamic pressure calculated using the $X$-component of velocity, $v_X$, divided by the solar wind dynamic pressure using the solar wind $v_X$. The black contour shows where this Plaschke criterion exceeds 0.25, and white where it exceeds 0.5, as defined in Plaschke et al. (2013).

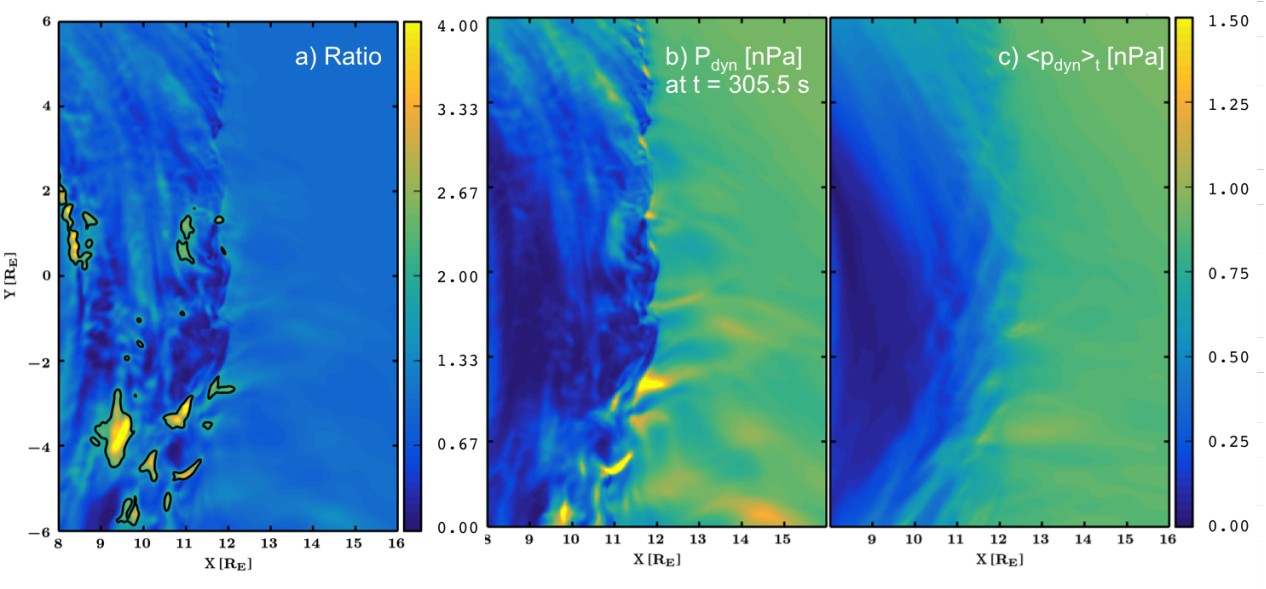

**Figure 3.** a) The Archer and Horbury (2013) criterion defined in Eq. (2), devised from the ratio of b) the dynamic pressure and c) the temporal average of dynamic pressure over three minutes centered at the time showing the jet-like feature in Fig. 1a. Panel a) shows a contour marking the locations where the ratio of panel b) and c) exceeds 2. Panels b) and c) have the same scale, from 0 to 1.5 nanopascals.

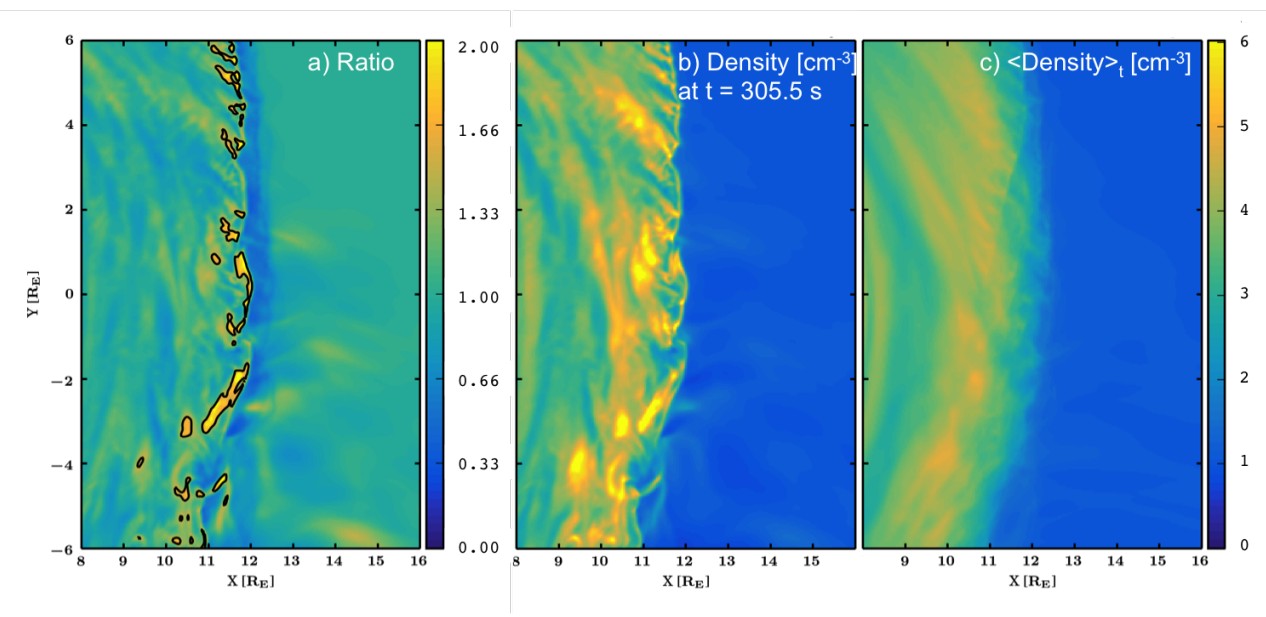

**Figure 4.** a) The Karlsson criterion in Eq. (3) (Karlsson et al., 2012, 2015), devised from the ratio of b) the density at $t = 305.5$ s, and the c) temporal average of density over three minutes, centered at the time showing the jet-like feature in Fig. 1a. Panel a) shows a contour marking locations where the ratio of panel b) and c) exceeds 1.5. Panels b) and c) have the same scale, from 0 to 6 particles in a cubic centimetre.

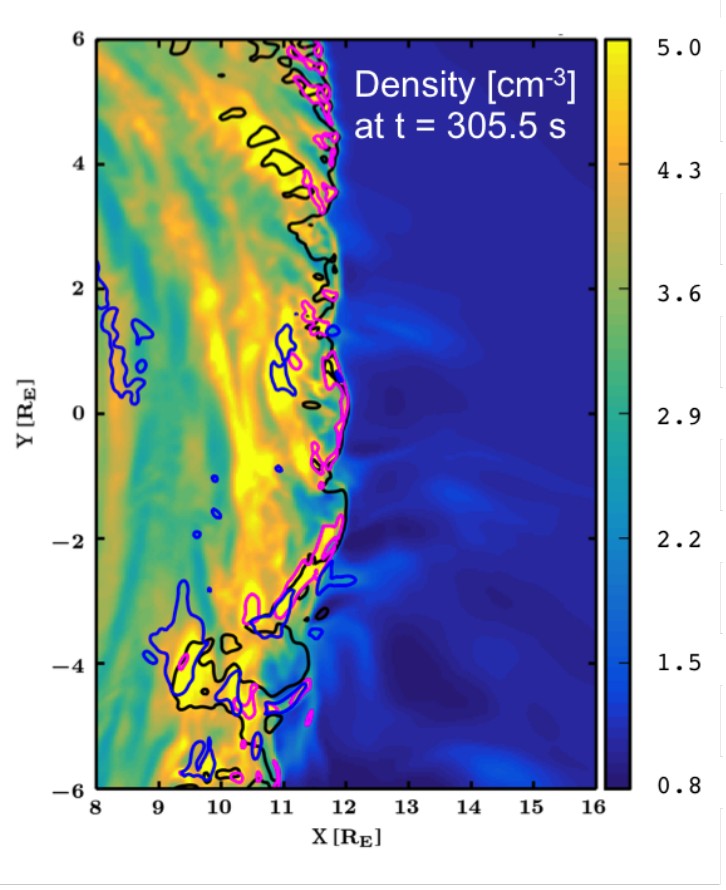

**Figure 5.** All criteria with density colour-coded at $t = 305.5$ s. The Karlsson criterion $C_K$ in Eq. (3) is given with magenta, Archer and Horbury criterion $C_A$ in Eq. (2) with blue, and the Plaschke criterion $C_P$ in Eq. (1) with black.

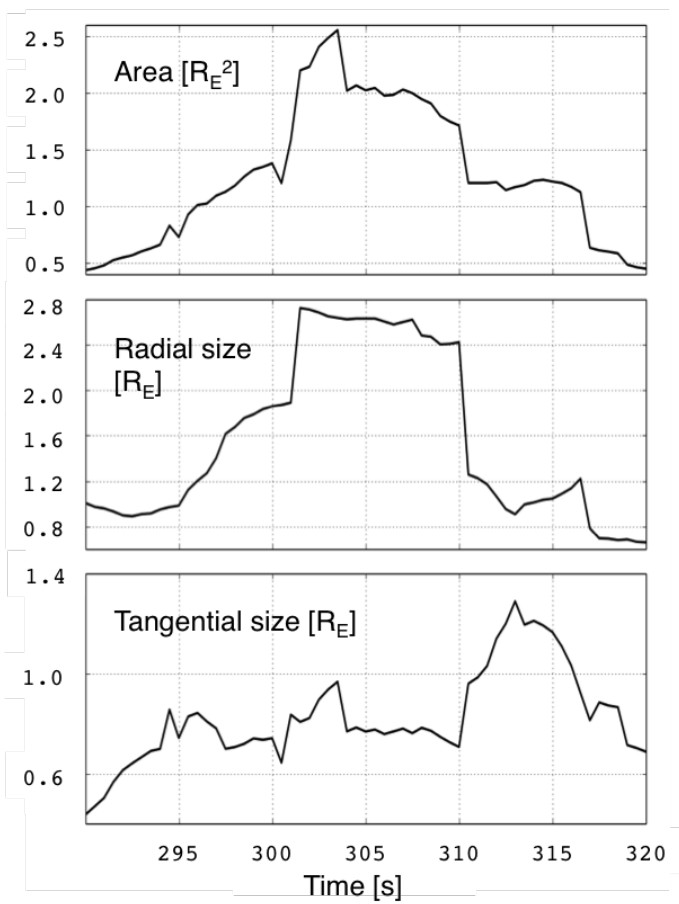

**Figure 6.** The jet area, radial, and tangential size as a function of time. The area has been calculated based on both the Archer and Horbury as well as the Plaschke criteria, while the radial size is the subtraction of the maximum and minimum radial distance of the jet boundary positions, reflecting the jet maximum extent. The tangential size is the effective jet width, and it is calculated by dividing the jet area by the radial size.

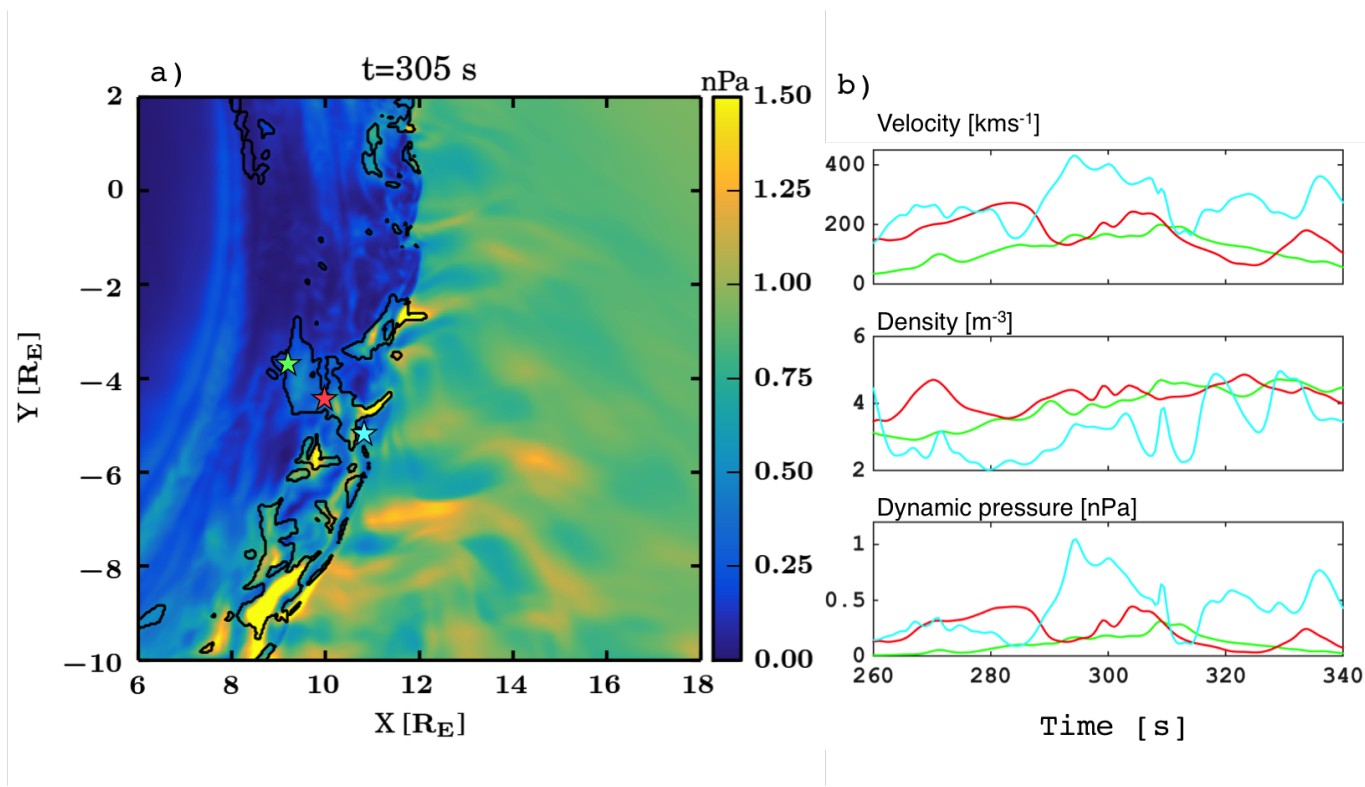

**Figure 7.** Jet evolution in time as a function of distance from the bow shock. a) An overview plot of the dynamic pressure with the Plaschke criterion with black contour, at time $t = 305$ s. The panel a) shows three locations with a green, red and cyan stars, at which virtual spacecraft data are given in panel b), showing from top to bottom the velocity, density and dynamic pressure against time. Colour coding shows the data from the similarly coloured stars in panel a).

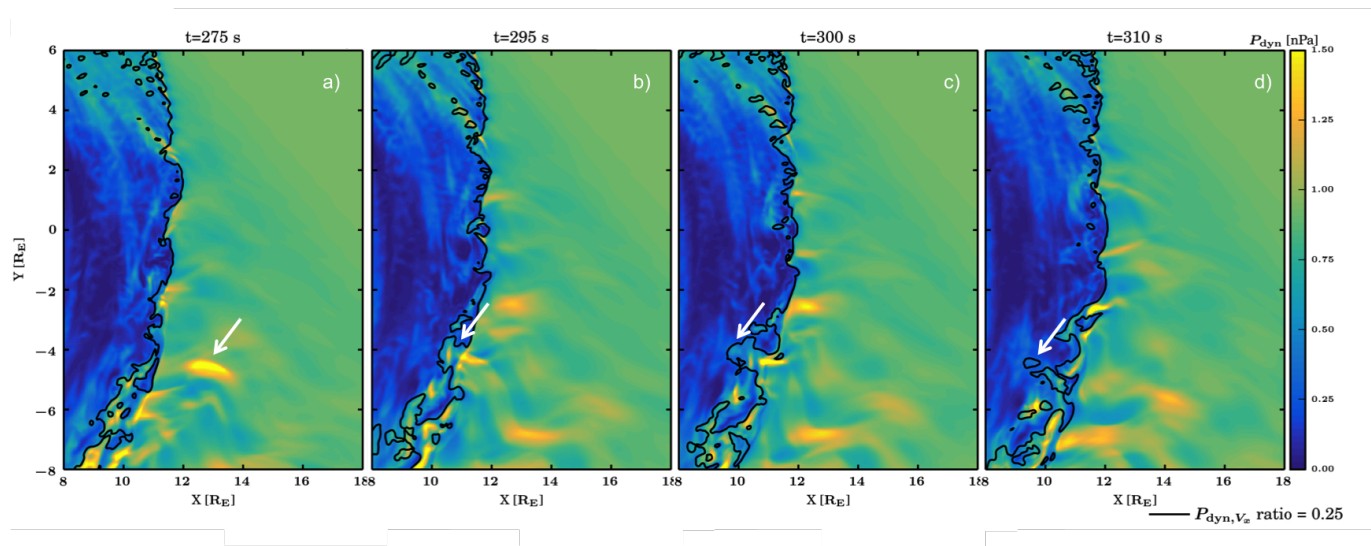

**Figure 8.** Time evolution of the jet. The colour coding in the background shows the total dynamic pressure, while the black contour shows the Plaschke criterion $C_P$ computed using the $X$ component of the velocity $v_X$ in dynamic pressure. Panels a) to d) show times 275, 295, 300, and 310 seconds, respectively, from the start of the simulation. The time of the jet at its prime is shown in Fig. 5. The white arrows show the jet generation and are referred to in the text. The panels are snapshots of Supplementary movie S2.

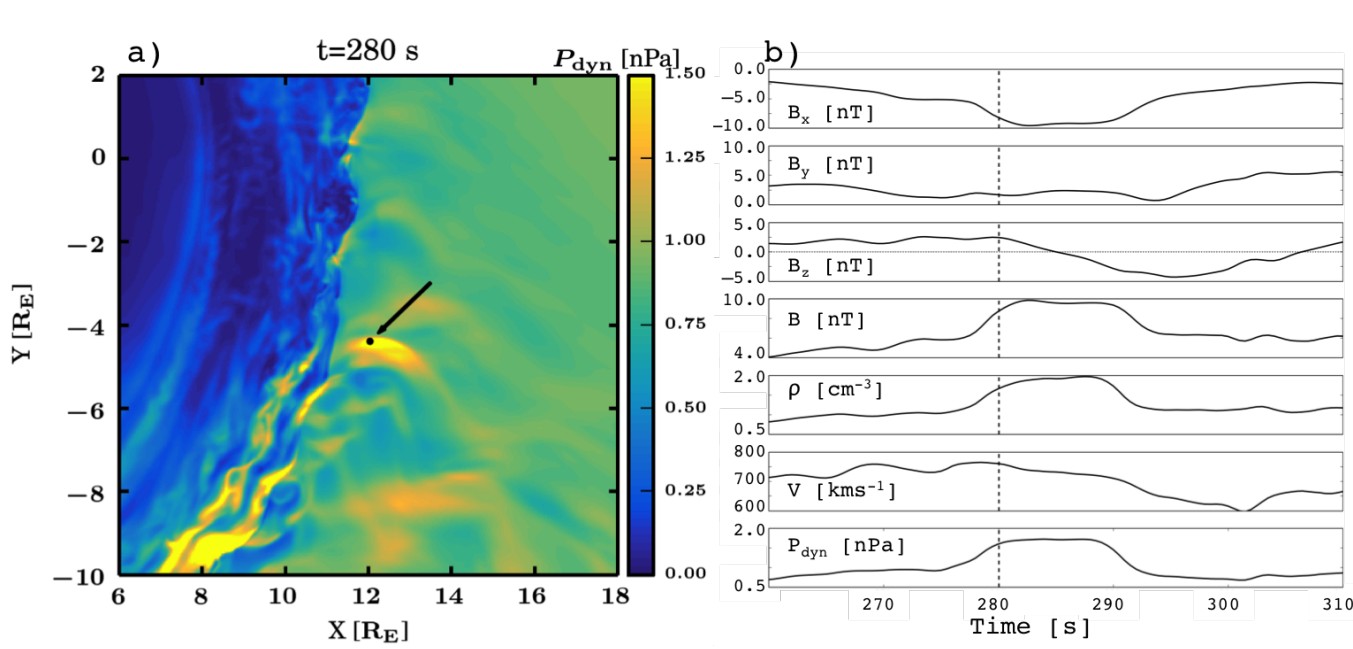

**Figure 9.** a) An overview plot of the high-pressure structure that causes the jet, colour-coding shows the dynamic pressure. The black dot marked by the arrow shows the virtual spacecraft location, for which different parameters are shown in panel b). From top to bottom the virtual spacecraft parameters are $X$, $Y$, and $Z$ components of the magnetic field, magnetic field intensity $B$, density $\rho$, total speed $v$ and the dynamic pressure $p_{dyn}$. The parameters are plotted against time, and the time shown in the panel a) is given by a dashed vertical line.