# Peer review of "Magnetosheath jet properties and evolution as determined by a global hybrid-Vlasov simulation"

_Annales Geophysicae, 2018_

## Referee Comment (RC1) · Anonymous Referee #1 · 15 Apr 2018

The authors used a global hybrid-Vlasov simulation to study magnetosheath jets. They identified one magnetosheath jet that satisfies all the selection criteria of Plaschke et al. (2013), Archer and Horbury (2013), and Karlsson et al. (2015). They conclude that the size of magnetosheath jet is ∼2.3 x 0.5 Re and the jet is generated because of an interaction of the foreshock ULF waves and the bow shock surface. These conclusions are neither substantial nor supported by the provided evidence. Therefore, the referee cannot recommend its publication in AG. Detailed comments are listed below:

1. With just a short description about the foreshock ULF waves in the Discussion, it is difficult to understand how the high dynamic pressure is associated with the waves. The authors are required to do a detailed analysis, like what they did for an identification and validation of the magnetosheath jet.

[Figure]

2. The authors chose 1 cmˆ-3 and 750 km/s for the solar wind density and velocity. The equivalent dynamic pressure is 0.94 nPa, which is considered a special solar wind condition. The authors need to explain why they chose such a condition. Can the magnetosheath only be seen in this condition or any other condition? Readers will be interesting in knowing about it.

3. A dynamic pressure of 0.94 nPa results in the subsolar magnetopause standoff distance of 11.5 Re. But the standoff distance derived from the model is about 7 Re, as seen in Figure 1. The same problem occurs for the position of the bow shock. From the movies S1 and S2, the bow shock is gradually expanding and the magnetopause is gradually shrinking. The locations of the bow shock and magnetopause never reach a steady state. This problem has made the referee think that this hybrid Vlasov model might not stable, giving unrealistic positions of the bow shock and magnetopause. To a validation of the hybrid Vlasov model, the authors are strongly suggested to add the locations of the bow shock and magnetopause to their simulation results using an empirical model.

4. The definition of a magnetosheath jet is a bit confusing. In my opinion, it should go with a criterion of flow speed, but the selection criteria of Plaschke et al. (2013), Archer and Horbury (2013), and Karlsson et al. (2015) are all related with the dynamic pressure or density. The authors need to classify this issue and add a definition of the magnetosheath jets to the beginning of the Introduction.

5. In Figure 3a, it shows that the geometry of the magnetosheath jets by Archer and Horbury (2013) is well aligned with the surface of the magnetopause. Are they really jets? The jets found by Karlsson et al. (2015), as shown in Figure 4a, look tiny and sporadic. Are they really jets? The features, which are shown in Figure 2 by Plaschke et al. (2013), are jets-like. But these jets never touch the surface of the magnetopause, which is different from the results by Plaschke et al. (2016).

6. The X and Y scales in Figures 3, 4, and 5 should be the same.
In summary, only one conclusion about the size of the magnetosheath jet is not substantial for a publication in AG. The authors are required to add more conclusions, such as a proof on the association between the high dynamic pressure and the foreshock ULF waves (Item 1), and the solar wind condition for an occurrence of the jet (Item 2). In addition, the authors are required to clarify the potential problem in their model (Item 3) and the issue in the definition and features of the jets (Items 4 and 5).

---

## Referee Comment (RC2) · Anonymous Referee #2 · 20 Apr 2018

Summary: This manuscript examines the physics of magnetosheath jets, using the results of a 5D vlasov simulation of the solar wind – magnetosphere interaction performed using the Vlasiator code. The simulation set up uses steady solar wind conditions, and several magnetosheath jets are reported to occur. The manuscript provides a detailed analysis of one jet in particular that is relatively large, and examines how three different identification criteria, previously published, capture the structure. The size of the jet is quantified, and is found to be consistent with experimental observations. Finally, the magnetosheath jet is shown to be associated with a variation in the upstream pressure that is caused by foreshock waves. The work provides a useful counterpoint to observational studies by showing for the first time that the different signatures adopted in different studies can in fact identify the same event, and therefore

help to unify understanding of what these structures are. It also provides a global view of the phenomenon, and contact is made with observations by estimating the size of the jet.

Overall, my primary concern with the manuscript is that it does not do full justice to what is a very interesting and important simulation result. It is important to compare the three identification criteria, but I think there is more that should be done. This relates to the physics questions about how the jets are formed and their impact on the magnetopause which will be of wider interest. I would be unwilling to recommend the manuscript for publication without addressing the following two points:

1) There is some limited discussion about the source of the magnetosheath jet, but the Vlasiator data surely allows for a much more detailed examination of the proposed formation mechanism and the nature of the ULF waves. In particular, it should be possible to generate some virtual spacecraft data for the upstream ULF waves (e.g. placed just upstream of the shock from where the jet arises) and see immediately if it is the formation of a SLAMS that happens here. Showing and discussing the data would significantly strengthen the manuscript. Similarly, how does the profile of the shock change as the ULF wave pressure front arrives and the jet begins to penetrate into the magnetosheath? Providing more information about the formation mechanism would significantly strengthen the paper and I think it would not be too difficult to extract this information.

2) I was surprised that there is no discussion about the impact of the jet on the magnetopause. In supplementary movie 1, at around t = 325 − 340 s, there is an oscillation of the magnetopause at x = 7.5, y = -4 (very roughly) which seems to follow directly from the arrival of the remnant of the jet. Two pulses traveling away from the impact point along the magnetopause are visible, and I wonder if this is reconnection triggered by the jet. Again I think it would significantly strengthen the paper to add information about the fate of the jet and its impact on the magnetopause.

Each of these would probably require more than one figure and the addition of several paragraphs of text or a section in the manuscript.

Further comments on the manuscript:

3) Evolution of the jet size. The jet size is quoted for one particular time, but it would be very good to provide more information about how the jet size changes. In particular, does the length parallel to the flow change more than the length perpendicular? This should be possible to extract from the simulation as well.

4) Jet occurrence. Watching the movies in the supplementary information, it seems that other jets do occur. Given the fact that the simulation is scaled to the Earth, is it possible to say anything about the occurrence rate and if this is consistent with observations?

5) Change in jet profile. It would be very useful from a spacecraft observation point of view to know how the profile of the jet - as would be observed by the spacecraft – changes with distance from the shock. Again this is something that Vlasiator would be able to show more clearly, and would be able to be extracted from the data.

---

## Short Comment (SC1) · 27 Apr 2018

The authors simulated magnetosheath jets using the VLASIATOR global hybrid-Vlasov solver. Palmroth et al. applied different methods to identify the jets. The team determined the size of the jets and . . . that is all.

Why did you do this simulation? What questions did you want to answer? What new physical processes did you discover? Do the spacecraft observations confirm the properties and evolution of the jets? Did you compare the VLASIATOR output to hybrid simulation results? What is the new result that the new hybrid-Vlasov solver provides?

---

## Short Comment (SC2) · 28 Apr 2018

I have just missed the deadline of short comments for the "Blanco-Cano et al., Cavitons and spontaneous hot flow anomalies in a hybrid-Vlasov global magnetospheric simulation" manuscript. The authors of this paper are almost similar to the SHFA related paper hence I leave my comments here. Sorry for the mess.

—

This is a very nice paper about the development of the foreshock cavitons and magnetosheath cavities based on unique real size global hybrid-Vlasov simulations. However, the simulation accuracy is questionable and the simulation results cannot reproduce the main features of the spontaneous hot flow anomalies (SHFA): the "shoulders" of

the cavity (the shocks around the SHFA) and the considerable drop of the SHFA. (The latter feature gave the name of the phenomenon!!!)

I would like to help the authors to prepare the next version of the manuscript. Hence I give my main concerns and comments below:

Title: I suggest using "Temporal development of foreshock cavitons in a hybrid-Vlasov global magnetospheric simulation". These phenomena are not SHFAs at all.

Page 2, Line 19-20: "[. . .] hot flow anomalies (HFAs) (Schwartz et al., 1985; Schwartz, 1995), [. . .]"

Facsko et al. (2010) wrote a review paper about HFAs. It would be appropriate to add to the list.

Page 2, Line 35: "The formation of an HFA needs an external perturbation in the solar wind, e.g., a current sheet interacting with a bow shock."

Actually the current sheet must be oriented appropriately and there are conditions for the rotation of the magnetic field in the discontinuity (Schwartz et al., 2000; Facsko et al., 2008, 2009, 2010). Both HFAs and SHFAs prefer high solar wind speed conditions (Safrankova et al., 2002; Facsko et al. 2008, 2009, 2010)

Safrankova, J. et al., The structure of hot flow anomalies in the magnetosheath, Advances in Space Research, Volume 30, pages 2737-2744, 2002

Schwartz, S. J. et al., Conditions for the formation of hot flow anomalies at Earth's bow shock, Journal of Geophysical Research, Volume 105, pages 12639-12650, doi: 10.1029/1999JA000320, 2000

Page 3, Line 5: "The proposed formation mechanism for SHFAs includes multiple ion reflections between foreshock cavitons and the bow shock (Omidi et al., 2013), as cavitons approach the shock, and ion trapping occurs in the cavitons."

Similar mechanism heats the HFAs too. However, the convective electric field focuses

and leads back the accelerated and back-scattered ions to the bow shock.

Page 3, Line 13-15: "As a consequence of ULF waves, shocklets and SLAMS merging into the shock, the quasi-parallel portion of the bow shock is far from being a single well defined surface, but instead forms a highly corrugated/rippled extended structure, where inhomogeneous heating and solar wind processing can take place (see, for example, Schwartz and Burgess, 15 1991; Omidi et al., 2005; Blanco-Cano et al., 2009)."

. . . and very strong acceleration processes as well (Wilson et al., 2016).

Wilson, L. B. et al., Relativistic Electrons Produced by Foreshock Disturbances Observed Upstream of Earth's Bow Shock, Physical Review Letters, Vol. 117, No. 21, doi:10.1103/PhysRevLett.117.215101, 2016

Page 4, Line 19-24: These are not typical solar wind parameters and not typical parameters for HFA/SHFA formation. Why did you choose such high solar wind speed? Why is the solar wind density so low? (Lower solar wind density is normal at HFA formation according to Facsko et al. (2009), Figure 11a). What are the components of the IMF? Do you have a Bx component?

The simulation time is quite short. The inbound solar wind at T=0s reaches ∼166 RE until the end of the simulation. Is it enough for reaching a quasi-stationary state? How did you do the initialisation of the simulation? Could you please present the video of the full simulation domain?

Page 5, Line 33-34: "SHFAs are also characterized by decrements in density and magnetic field strength, but have in addition a higher temperature than the surrounding plasma."

And the surrounding shock and the huge solar wind velocity depletion. These features cannot be neglected.

Page 5, Line 34-Page 7, Line 1: "However, setting a criterion on the temperature is

not straightforward since SHFAs are immersed in the foreshock, which has a higher temperature than the pristine solar wind."

Where have you observed several 10 MK temperature in the foreshock?

Page 7, Line 1-3: "[...] deviations from the bulk solar wind velocity are observed throughout the foreshock, and they are not prominent enough inside SHFAs to be unambiguously identified."

Those phenomena that do not show anomalous flow cannot be called Spontaneous Hot FLOW ANOMALY. (And neither are they HOT.)

Page 6, Figure 1: Could you please provide an image of the full simulation domain with the same colours and scale? Has the VLASIATOR simulation reached a quasi-stationary state?

Page 7, Line 3-6: Provide a reference for this method or prove that it is applicable here.

Page 7, Line 13-14: "In a 3-D run, the total number of these structures in the whole foreshock would most likely be larger."

Why?

Page 8, Line 3-31: Where is the cavity?

Page 8, Line 33-34: "The shaded areas in Figure 3 show how at time T2 there are multiple large cavitons upstream of the shock. SHFAs are found at shaded areas where also beta > 10."

I see neither SHFA/cavity nor velocity depletion on Figure 3. This event cannot be a SHFA.

Page 10, Figure 3: I do not see any SHFA here. The density and the magnetic field should drop significantly. The temperature should reach several 10 MK. The solar wind flow should drop significantly.

Page 10, Line 1-2: "In the sixth panel, small dips in the value of |V| associated with cavitons and SHFAs can also be identified."

I cannot see any dips, only the bow shock transition is visible. Actually the solar wind speed should drop significantly, a few 100 km/s.

Page 12, Line 26/27: "Position C is located within an upstream SHFA [...]"

Page 14, Figure 6c, h, n: The (S)HFAs are formed by the interaction of the solar wind ions and the reflected and accelerated ion beam of the bow shock. In young (S)HFAs the two populations can be distinguished by the ion velocity distributions at $V_x=0$ km/s and $V_x=600$ km/s. (Lucek et al, 2004, Figure 4b; Zhang et al., 2010, Figure 7b). These events must be young (at least C). I cannot see the typical distribution with double peak. Hence these structures are not SHFAs or they aged very quickly. (The mature (S)HFAs have no such velocity distributions.)

Page 16, Figure 7: See comments for Figure 3.

Page 17, Line 26: "We suggest that suprathermal beam ion PADs are a useful tool for identifying SHFAs in the foreshock."

. . . according to Kecskemety et al. (2006).

Page 18, Figure 8: The distribution of the relative amount (count/total count) would be compared easier.

Page 20, Figure 9: The drop of the density and magnetic field is not sufficient. The depth of the drop neither. The temperature should reach several million K even in a SHFA. I cannot see any velocity drop here.

Page 20, Line 14-15: "This may also explain why the caviton shown in Fig. 9 does not display the "shoulders" of enhanced plasma density identified in spacecraft observations on either sides of the density and magnetic field depression."

These "shoulders", the shocks are very important features of the SHFAs. Their presence proves that the cavity is not in equilibrium and expands. If the VLASIATOR cannot create them that is big a problem. The hybrid simulations of Nick Omidi and Yu Lin could present these shoulders. What is the advantage of using VLASIATOR if the hybrid-Vlasov code cannot present these shocks? Furthermore, these shocks lead to the observed depletion of the solar wind velocity because the deceleration of the solar wind comes from the bad fitting and plasma moment calculation (Kecskemety et al., 2006, Figure 3 and 7; Parks et al., 2013).

Page 22, Line 19-20: "That is, the flow of the thermal solar wind core was not slowed or deflected, but rather, changes in bulk flow are due to the combination of a density decrease for the core and a strengthening of the suprathermal beam. When the thermal core is depleted, the backstreaming beam can have a relatively greater impact on bulk velocity measurements."

In this case the decrease of the velocity would be deeper. See my comments above, the lack of "shoulders" in density and magnetic field is related to the missing velocity decrease. This is an important feature that cannot be neglected. If these features are missing, the phenomena are not SHFA or the VLASIATOR needs further improvement to be able to study them.

---

## Author Comment (AC1) · 9 May 2018

Helsinki, May 9, 2018

Dear Referee #1,

Thank you for your thorough review of our paper. Below, we go through the points in detail; the original Referee questions are marked with italics.

*The authors used a global hybrid-Vlasov simulation to study magnetosheath jets. They identified one magnetosheath jet that satisfies all the selection criteria of Plaschke et al. (2013), Archer and Horbury (2013), and Karlsson et al. (2015). They conclude that the size of magnetosheath jet is ~2.3 x 0.5 Re and the jet is generated because of an interaction of the foreshock ULF waves and the bow shock surface. These conclusions are neither substantial nor supported by the provided evidence. Therefore, the referee cannot recommend its publication in AG.*

We would like to maintain that the size of the jet and its generation are important results warranting publication. Even with multi-spacecraft data, the scale-size observations have been indirect and inferred, based on statistics rather than individual structures. We think it is important to establish with a model that there is indeed a coherent structure, with generation and decay, whose size is in agreement with the interpretation of spacecraft data. Since this has not been done before, it is important to first do this rigorously and compare to different observational criteria, leading to a proof-of-concept that can then in the future be used more easily, without having to verify all different jet-like structures separately. The fact that we get similar scale sizes as compared to observations lends credibility to the observations as well, needed to properly interpret the observational studies so far. Further, the jet generation has not been verified with a model before. We further emphasize that the jet size and the generation mechanism are not the only results of the manuscript, as we also clarify how the different criteria in the literature are related, and verify that they can occur for steady IMF.

*Detailed comments:*

*1. With just a short description about the foreshock ULF waves in the Discussion, it is difficult to understand how the high dynamic pressure is associated with the waves. The authors are required to do a detailed analysis, like what they did for an identification and validation of the magnetosheath jet.*

We agree with this, and thank the Reviewer for pointing this out. Should the Editor decide to ask for a revision of the manuscript, we will add a detailed description of the upstream structure that caused the jet.

*2. The authors chose 1 cmˆ-3 and 750 km/s for the solar wind density and velocity. The equivalent dynamic pressure is 0.94 nPa, which is considered a special solar wind condition. The authors need to explain why they chose such a condition. Can the magnetosheath only be seen in this condition or any other condition? Readers will be interesting in knowing about it.*

This is due to the run conditions we originally chose. Vlasiator is a supercomputing code requiring a large computer to be run, and therefore for each run we need to separately ask for resources from different supercomputing centres. The 750 km/s is originally chosen because we have needed the solar wind to flush through the simulation box rather quickly so that the initialized magnetosphere appears without too much waiting. The density is chosen such that the combination of the density and velocity yields such an Alfvén Mach number that the foreshock will be representative of the reality. Thus we can trust the foreshock physics and consequently its bow shock interactions. The magnetosheath appears in our other runs as well, and has been verified in other peer-reviewed

papers to represent reality, e.g., Hoilijoki et al., 2016 JGR.

A dynamic pressure of <= 0.94 nPa occurs 16% of the time throughout the solar cycle and 23% of the time under quasi-radial IMF, based on OMNI solar wind data for the last solar cycle. While our case does not represent the median conditions, it is not an outlier. A recently accepted review paper by Plaschke et al. (2018) states that observational statistics show a slight tendency in the jet occurrence for higher solar wind speeds and lower densities than usual. Full statistics of jet occurrence with conditions are not possible to be carried out with Vlasiator due to the computational demand, but may be possible with a limited number of runs. We are enthusiastic that such statistics could be carried out, however, first we need a detailed comparison of the different jet criteria so that we can run such a statistical study in practice.

*3. A dynamic pressure of 0.94 nPa results in the subsolar magnetopause standoff distance of 11.5 Re. But the standoff distance derived from the model is about 7 Re, as seen in Figure 1. The same problem occurs for the position of the bow shock. From the movies S1 and S2, the bow shock is gradually expanding and the magnetopause is gradually shrinking. The locations of the bow shock and magnetopause never reach a steady state. This problem has made the referee think that this hybrid Vlasov model might not stable, giving unrealistic positions of the bow shock and magnetopause. To a validation of the hybrid Vlasov model, the authors are strongly suggested to add the locations of the bow shock and magnetopause to their simulation results using an empirical model.*

Thank you for making this comment, this is very helpful indeed.

Regarding the expansion of the magnetopause and the bow shock, we would like to point out that in all hybrid-kinetic simulations, hybrid-PIC included, there is a gradual increase of the bow shock position for two reasons. First is the magnetic field pile-up due to the 2D setup of the run. The field piles up at the magnetopause because it cannot slip towards the nightside as in reality. We emphasize that this is a feature in all hybrid-kinetic simulations, and there is not much one can do about it. There are several other peer-reviewed papers showing this feature, indicating that it should not be regarded as a showstopper. Second, and smaller issue in our case is the artificial heating in the hybrid-kinetic simulations due to numerical diffusion. We have managed to develop such a good solver that the numerical heating stays at a tolerable level and does not largely contribute to the gradual expansion.

The simulation is initialized with the geomagnetic dipole field and the IMF pervading the box, while the plasma flows with the solar wind parameters. This causes the magnetosphere and bow shock structures to develop self-consistently during the initialization of the run. Regarding the magnetopause, we have looked more closely at the magnetopause position in Figure 1. First of all, in simulations (3D included), the magnetopause position is determined by 1) gradient of density, magnetic field or current density, 2) last closed field line, or 3) the so-called fluopause method introduced by Palmroth et al., 2003 (JGR). These parameters often do not agree with each other, while it has been shown that the fluopause gives the closest agreement with empirical models, as shown e.g., in Palmroth et al. 2003. The gradients of the abovementioned parameters vary between 2 Re in Figure 1, while the fluopause method puts the subsolar magnetopause into about 10 Re. The density enhancements that are shown closer to the earth, at about 7 Re are due to the pile-up, and they are not related to the magnetopause according to the above criteria. They originate from plasma that has been brought there before and is being squeezed by the new incoming plasma.

If a revision is decided, we will explain these issues in the manuscript and add the magnetopause position to the plots to guide the eye.

*4. The definition of a magnetosheath jet is a bit confusing. In my opinion, it should go with a criterion of flow speed, but the selection criteria of Plaschke et al. (2013), Archer and Horbury (2013), and Karlsson et al. (2015) are all related with the dynamic pressure or density. The authors need to classify this issue and add a definition of the magnetosheath jets to the beginning of the Introduction.*

The Reviewer is right in that especially the early observations are more related to the flow speed, while especially in the later years the vast majority of previous studies have used dynamic pressure and not velocity as the key quantity. However, since the flow speed appears quadratically in the dynamic pressure, it is also strongly reflected in the used criteria. In the revision, we will classify this in more detail in the Introduction, as the Reviewer suggests.

*5. In Figure 3a, it shows that the geometry of the magnetosheath jets by Archer and Horbury (2013) is well aligned with the surface of the magnetopause. Are they really jets? The jets found by Karlsson et al. (2015), as shown in Figure 4a, look tiny and sporadic. Are they really jets? The features, which are shown in Figure 2 by Plaschke et al. (2013), are jets-like. But these jets never touch the surface of the magnetopause, which is different from the results by Plaschke et al. (2016).*

We would like to point here that the observational community has adopted the term "jet", which has a connotation of an elongated feature. However, without proper modelling of them, we cannot actually say, based on the observations, what their dimensions are and what is their time evolution. It is true that some of them reach the magnetopause (while others probably do not, we cannot say this based on observations, either), indicating that at least some of them could be elongated features. However, observations nearer the shock have not estimated the sizes/shapes of jets, and therefore they may be more like "blobs" there. It is exactly the simulations that allow more detailed comparison of some of their properties (like size/shape and how large a fraction of them reach the magnetopause) that observations may be limited in inferring.

We also note that there is a magnetopause effect caused by the jet, visible in the S1 movie. We omitted this discussion because we tend to avoid making conclusions at the magnetopause due to the pileup effect. The other Reviewer urged us to add this in the manuscript and describe the magnetopause effect as well, which we shall do pending decision from the Editor.

*6. The X and Y scales in Figures 3, 4, and 5 should be the same.*

Will be corrected.

*In summary, only one conclusion about the size of the magnetosheath jet is not substantial for a publication in AG. The authors are required to add more conclusions, such as a proof on the association between the high dynamic pressure and the foreshock ULF waves (Item 1), and the solar wind condition for an occurrence of the jet (Item 2). In addition, the authors are required to clarify the potential problem in their model (Item 3) and the issue in the definition and features of the jets (Items 4 and 5).*

These will be clarified in the revision according to the above answers.

Thank you again for your very helpful and constructive comments, which will significantly increase the quality of the manuscript, we appreciate the time you spent on our work.

On behalf of all the co-authors,
Minna Palmroth

---

## Author Comment (AC2) · 9 May 2018

Helsinki, May 9, 2018

Dear Referee #2,

Thank you for your thorough review of our paper. Below, we go through the points in detail; the original Referee questions are marked with italics.

*Summary: This manuscript examines the physics of magnetosheath jets, using the results of a 5D vlasov simulation of the solar wind – magnetosphere interaction performed using the Vlasiator code. The simulation set up uses steady solar wind conditions, and several magnetosheath jets are reported to occur. The manuscript provides a detailed analysis of one jet in particular that is relatively large, and examines how three different identification criteria, previously published, capture the structure. The size of the jet is quantified, and is found to be consistent with experimental observations. Finally, the magnetosheath jet is shown to be associated with a variation in the upstream pressure that is caused by foreshock waves. The work provides a useful counterpoint to observational studies by showing for the first time that the different signatures adopted in different studies can in fact identify the same event, and therefore help to unify understanding of what these structures are. It also provides a global view of the phenomenon, and contact is made with observations by estimating the size of the jet.*

*Overall, my primary concern with the manuscript is that it does not do full justice to what is a very interesting and important simulation result. It is important to compare the three identification criteria, but I think there is more that should be done. This relates to the physics questions about how the jets are formed and their impact on the magnetopause, which will be of wider interest. I would be unwilling to recommend the manuscript for publication without addressing the following two points:*

*1) There is some limited discussion about the source of the magnetosheath jet, but the Vlasiator data surely allows for a much more detailed examination of the proposed formation mechanism and the nature of the ULF waves. In particular, it should be possible to generate some virtual spacecraft data for the upstream ULF waves (e.g. placed just upstream of the shock from where the jet arises) and see immediately if it is the formation of a SLAMS that happens here. Showing and discussing the data would significantly strengthen the manuscript. Similarly, how does the profile of the shock change as the ULF wave pressure front arrives and the jet begins to penetrate into the magnetosheath? Providing more information about the formation mechanism would significantly strengthen the paper and I think it would not be too difficult to extract this information.*

We fully agree with the Reviewer, and note that also the other Reviewer made this same point. This is indeed easy to add to the manuscript, and should this manuscript be accepted for revision, we shall carry out a detailed examination of the structure that causes our jet, along the lines that the Reviewer suggested.

*2) I was surprised that there is no discussion about the impact of the jet on the magnetopause. In supplementary movie 1, at around t = 325 – 340 s, there is an oscillation of the magnetopause at x = 7.5, y = -4 (very roughly) which seems to follow directly from the arrival of the remnant of the jet. Two pulses traveling away from the impact point along the magnetopause are visible, and I wonder if this is reconnection triggered by the jet. Again I think it would significantly strengthen the paper to add information about the fate of the jet and its impact on the magnetopause.*

Again, the Reviewer is absolutely right. We omitted this discussion because we tend to avoid making conclusions at the magnetopause due to the pileup effect (see our answer to the other Reviewer, point #3). We agree with the Reviewer and think that the magnetopause oscillation is

caused by the jet. We shall add this information and a related analysis to the manuscript, along with a proper discussion about the pileup effect.

*3) Evolution of the jet size. The jet size is quoted for one particular time, but it would be very good to provide more information about how the jet size changes. In particular, does the length parallel to the flow change more than the length perpendicular? This should be possible to extract from the simulation as well.*

Should the Editor accept this manuscript for revision, we shall add this information into the manuscript.

*4) Jet occurrence. Watching the movies in the supplementary information, it seems that other jets do occur. Given the fact that the simulation is scaled to the Earth, is it possible to say anything about the occurrence rate and if this is consistent with observations?*

Yes, indeed it is. However, we chose not to do this in this manuscript. This is because we would first like a proof-of-concept paper, where we verify the methodology, so that we can trust the results. Once this has been carried out, we can adopt the methodology to all our runs, to all our jets, leading to possibly (tens of) thousands of observation points in space and time, given that we have now several runs with varying conditions that can be used. It would be impractical to verify this many jets in this detail in practice, and therefore we thought it is good to verify one first.

*5) Change in jet profile. It would be very useful from a spacecraft observation point of view to know how the profile of the jet - as would be observed by the spacecraft – changes with distance from the shock. Again this is something that Vlasiator would be able to show more clearly, and would be able to be extracted from the data.*

This is an excellent suggestion. Both the Karlsson, and Archer and Horbury criteria are determined from the peak values, and the full-width-at-half-maximum approximation when analyzing the spatial scales. What indeed we could do, and we thank the Reviewer for pointing this out, is an evolution of the full-width-at-half-maximum parameter in time and space. We shall add this, if the Editor asks for a revision

Thank you again for your very helpful and constructive comments, which will significantly increase the quality of the manuscript, we appreciate the time you spent on our work.

On behalf of all the co-authors,
Minna Palmroth

---

## Author Comment (AC3) · 9 May 2018

Dear Dr Facsko,

As for the title of your comment, according to Merriam-Webster dictionary, the definition of physics is "a science that deals with matter and energy and their interactions, by looking at a) processes and phenomena of a particular system, or b) the physical properties and composition of something". This paper looks at b) the physical properties (i.e., size) of a magnetosheath jet, something that has not been done before with a real-size kinetic magnetospheric simulation, and a) the process at which the jet is formed by foreshock – bow shock interactions, again something that has not been looked at with a real-size kinetic magnetospheric simulation before.

[Figure]

As for the questions in your comment, please see the submitted manuscript.

On behalf of all the co-authors, Minna Palmroth